# Exposure to Zika and chikungunya viruses impacts aspects of the vectorial capacity of *Aedes aegypti* and *Culex quinquefasciatus*

Mônica Crespo[1], Duschinka Guedes[1], Marcelo Paiva[1,2], Mariana Sobral[1], Elisama Helvecio[1], Rafael Alves[1], George Tadeu[3], Claudia Oliveira[1], Maria Alice Varjal Melo-Santos[1], Rosângela Barbosa[1], Constância Ayres[1] *

1 Departamento de Entomologia, Instituto Aggeu Magalhães, Fundação Oswaldo Cruz (FIOCRUZ-PE), Recife, Pernambuco, Brasil, 2 Núcleo de Ciências da Vida, Centro Acadêmico do Agreste, Universidade Federal de (UFPE), Caruaru, Pernambuco, Brasil, 3 Núcleo de Estatística e Geoprocessamento, Instituto Aggeu Magalhães, Fundação Oswaldo Cruz Pernambuco (FIOCRUZ-PE), Recife, Pernambuco, Brasil

* constancia.ayres@fiocruz.br

**Data Availability Statement:** All relevant data are within the manuscript and its Supporting Information files.

## Abstract

Zika (ZIKV) and chikungunya (CHIKV) are arboviruses that cause infections in humans and can cause clinical complications, representing a worldwide public health problem. *Aedes aegypti* is the primary vector of these pathogens and *Culex quinquefasciatus* may be a potential ZIKV vector. This study aimed to evaluate fecundity, fertility, survival, longevity, and blood feeding activity in *Ae. aegypti* after exposure to ZIKV and CHIKV and, in *Cx. quinquefasciatus* exposed to ZIKV. Three colonies were evaluated: AeCamp (*Ae. aegypti*—field), RecL (*Ae. aegypti*—laboratory) and CqSLab (*Cx. quinquefasciatus*—laboratory). Seven to 10 days-old females from these colonies were exposed to artificial blood feeding with CHIKV or ZIKV. CHIKV caused reduction in fecundity and fertility in AeCamp and reduction in survival and fertility in RecL. ZIKV impacted survival in RecL, fertility in AeCamp and, fecundity and fertility in CqSLab. Both viruses had no effect on blood feeding activity. These results show that CHIKV produces a higher biological cost in *Ae. aegypti*, compared to ZIKV, and ZIKV differently alters the biological performance in colonies of *Ae. aegypti* and *Cx. quinquefasciatus*. These results provide a better understanding over the processes of virus-vector interaction and can shed light on the complexity of arbovirus transmission.

## Introduction

Complex interactions between vectors and arboviruses determine the vectorial competence of a species and act in association with biotic and abiotic factors. This set composes the parameters of vectorial capacity, which is described as the potential of a species to transmit a pathogen. In this context, aspects of the vector's biological performance stand out, such as age [1], reproductive capacity, longevity, blood feeding behavior and population density, which are considered determinants in the transmission process [2]. Studies have already demonstrated that the arbovirus infection influences the reproductive capacity of females of different species [3–9].

**Funding:** C.F.J.A: Grants APQ-1608-2.13/15 and APQ-0085-2.13/16, and M.H.S.P, grant APQ-0725-2.13/17. Fundação de Amparo à Pesquisa do Estado de Pernambuco (FACEPE). The funders had no role in study design, data collection and analysis, decision to publish, or preparation of the manuscript.

**Competing interests:** The authors have declared that no competing interests exist.

Modifications in longevity, survival, and in blood meal activity have also been associated with arbovirus exposure or infection [4, 7, 8].

The Zika (ZIKV) and chikungunya (CHIKV) viruses are RNA arboviruses from the *Flaviviridae* and *Togaviridae* families, respectively, which have quickly spread in recent years to various parts of the world, including Brazil [10–12]. These pathogens are a major public health concern, as they cause infections in humans that can trigger neurological complications, such as the Guillain-Barré Syndrome [13] and the Congenital Zika Syndrome [14], as well as cardiac manifestations [15] and painful and disabling polyarthralgia [16].

Recife, a municipality located in the northeast region of Brazil, is known to have very favorable environmental conditions for mosquito reproduction and maintenance of high vector population densities. These conditions, when associated with vectorial capacity and presence of a susceptible human population, construct a scenario conducive to the rapid propagation of an arbovirus [17]. Additionally, the ability of the species *Ae. aegypti* and *Ae. albopictus* to transmit ZIKV [18–21] and CHIKV [22–24], is an important factor for the spread of these viruses in the Americas [11, 25]. Also, it should be considered that other highly anthropophilic local species may be involved in arbovirus transmission as secondary vectors [26]. *Cx. quinquefasciatus*, for example, is an abundant species in many parts of the world, including Brazil. However, most studies on its role in ZIKV transmission were only published as of 2016 [27–33].

In general, the information available on different aspects of the vectorial capacity of *Aedes aegypti* is not sufficient to explain the dynamics of arbovirus transmission, especially after pathogenic infections, considering that viruses modulate several parameters of the biological performance of vectors, and the mechanisms responsible for this modulation are also not well understood [9]. Studies that address the various aspects of the vectorial capacity of species involved in the transmission of arboviruses can contribute to the elucidation of fundamental questions for guidance of vector population control programs. Thus, the present study aimed to evaluate the effect of exposure to ZIKV and CHIKV on fecundity, fertility, survival, longevity and blood meal activity in *Ae. aegypti* from the city of Recife (Pernambuco, Brazil) and in a laboratory colony, as well as the effect of ZIKV on *Cx. quinquefasciatus*, a laboratory colony, considering the same aspects, which are relevant for arbovirus transmission [4, 7, 34–36].

## Materials and methods

### Study area

The study was carried out in Recife (8°03'S 34°52'W), capital of the state of Pernambuco, Brazil, located in the northeast region. In the dry season, rainfall is scarce in the region, but it can be intense in the rainy season (from April to July). The average annual rainfall in the city is 2155.5, according to the National Meteorology Institute (INMET), in the period from 1991 to 2020 [37]. The temperature and relative air humidity range from 22 to 32°C and from 70 to 90% at different times of the year [38]. Recife is an hyperendemic area, with multiple arboviruses circulating simultaneously, and it has been considered as the epicenter of the first Zika outbreak in Brazil [39].

### Mosquito samples

Three different mosquito populations were used in the present study: two laboratory colonies (RecL and CqLab) and one natural population (AeCamp). The laboratory colony of *Ae. aegypti* (RecL) has been maintained without contact with larvicide or adulticide since 1996, when it was established from collections performed in the neighborhood of Graças, in Recife [40]. The other *Ae. aegypti* population, called AeCamp, came from the field, and was established through egg collections performed between 2017 and 2018, in 13 neighborhoods in Recife (Santo

Amaro, Várzea, Afogados, Dois Irmãos, Apipucos, Monteiro, Ipsep, Boa Viagem, Nova Desco-berta, Vasco da Gama, Cidade Universitária and Mustardinha). Experiments with AeCamp were carried out with individuals of the $F_2$ generation. As for the *Culex* species, we have used the *Cx. quinquefasciatus* laboratory colony, known as CqSLab, which was originated in the municipalities of Ipojuca, Olinda and Jaboatão dos Guararapes (Metropolitan Region of Recife), and it has been maintained since 2011 [41].

Eggs were collected in the field using BR-OVT traps [38] and larvae and pupae were col-lected directly from the breeding sites, using a larval dipper. All colonies were kept in the insec-tary of the Entomology Department of the Instituto Aggeu Magalhães (IAM-FIOCRUZ-PE), under controlled conditions of temperature ($26˚C \pm 1˚C$), relative humidity (50 to 90%) and photoperiod (14:10 h—L/D). Larvae and pupae were kept in breeding containers with potable water and were fed cat food (friskies®). Adult mosquitoes were kept in aluminum mesh cages (50 x 40 cm) and were fed with a 10% sucrose solution *ad libitum* on a daily basis. For females, blood feeding was offered weekly, in an artificial feeder, using defibrinated rabbit blood (*Oryc-tolagus cuniculus)*.

## Virus strains

ZIKV (BRPE243/2015) [42] and CHIKV (PE480/2016) strains were obtained from patients residing in the State of Pernambuco, Brazil, during the 2015 and 2016 outbreaks, respectively. These viruses were kindly provided by Dr. Marli Tenório (Virology Laboratory, FIOCRUZ–PE). Viruses were propagated in Vero cells and titered as described in Guedes et al. [30].

## Artificial bloodmeal for virus infection

Two to three independent experiments were performed with ZIKV and CHIKV. Each experi-ment was performed with two groups for each colony (RecL, AeCamp and CqSLab): exposed to the virus (E) and non-exposed (NE, the control group). Two hundred adult females were used for the control groups (NE) and 300, for the test groups; they were aged between 7 and 10 days of emergence. In all groups, females were starved for 24 hours prior to oral exposure to the virus.

Oral blood-feeding was provided with an artificial feeder comprised of a Petri dish and Par-afilm®, with a mixture of cultures of Vero cells inoculated with virus and defibrinated rabbit blood, in a volume of 10 mL, in a 1:1 ratio, as described in Guedes et al. [30]. The stock virus dose used was $10^6$ and $10^9$ plaque forming units per ml (PFU/ml), for ZIKV and CHIKV, respectively. The negative control was a mixture of equal volume of virus-free cell culture and defibrinated rabbit blood. Approximately 0.5 ml of each mixture was aliquoted for further titration. Females were exposed to a blood meal for one hour. All females had three more blood meals devoid of virus, after exposure to the virus, with the objective of evaluating blood meal activity and keeping them in active gonotrophic cycles for a better evaluation of longev-ity. These blood meals were provided once a week for three consecutive weeks.

## Assessment of the biological parameters after exposure to viruses

For the analysis of the putative biological cost, groups were defined according to the results of exposure to the viruses. Thus, for ZIKV, groups were divided into three: not exposed (NE); exposed, but not infected (E) and exposed and infected (EI). This definition was possible, since all mosquitoes were processed individually, by RT-qPCR, to confirm the infection, following Guedes et al. protocol [30]. For CHIKV, groups were divided into two: not exposed (NE) and exposed and infected (EI), considering that, for this virus, infection rates were higher than ZIKV (above 90%), the number of exposed individuals and not infected was not enough for

statistical analysis (representing 4 and 8% for RecL and AeCamp, respectively), and therefore those samples were discarded.

Approximately 24 hours after the blood meal, 50 engorged *Ae. aegypti* females from each experimental group were transferred to individual cages. This was the initial number of samples used for all biological cost assessments.

## Fecundity and fertility assessment

On the 3$^{rd}$ day post-exposure (dpe), each cage received a container (measuring 3.14 cm$^3$) to mimic an oviposition site that contained 30% grass infusion and, in the case of the experiments with *Ae. aegypti*, posture supports made of cardboard, measuring approximately 4 x 2 cm.

In the *Ae. aegypti* assays, cardboards were changed twice a week, in three gonotrophic cycles. The collected eggs were counted through a stereomicroscope to determine the fecundity rate of each female. After 15 to 20 days of quiescence, the eggs were placed in recipients containing 2 mL of 30% grass infusion to stimulate the synchronous hatching of the larvae. These were counted to estimate the individual fertility rate of each female. Fecundity and fertility analyses were performed in the first gonotrophic cycle.

In the *Cx. quinquefasciatus* assays, the recipients were removed from the individual cages after each oviposition (once a week), in three gonotrophic cycles. The number of eggs present in each collected raft were determined with the aid of a magnifying glass, up to 24 hours after laying. Larvae hatching percentage was evaluated approximately 72 hours after oviposition, because the eggs of this species do not go into quiescence. Fecundity and fertility analyses were performed with all eggs from the first gonotrophic cycle.

## Survival and longevity assessment

Mortality notification was performed daily to assess survival and longevity. To detect viral infection, ten females in each experimental group of *Ae. aegypti*, were collected at three time points: 7, 14 and 21 dpe (days post exposure). There was a total of 20 females from each group until the end of life, which were also collected as they died throughout the experiment, to detect viral infection. For these, FTA—*classic card Whatman*® cards (cards), containing *Manuka honey blend* ® honey, were made available on the screens of the cages, from the 7th to the 14th dpe. All females contributed to survival, and longevity assessments up to the point at which they left the study. For the analysis of average lifetime (longevity), females collected on the 7th, 14th and 21st dpe were excluded.

For *Cx. quinquefasciatus*, we have collected females only when death occurred throughout the experiments. This was necessary to guarantee a minimum number of infected females for analysis of longevity and survival in this species, considering that data from experiments carried out by our group [30] and unpublished data show a lower rate of ZIKV infection in *Cx. quinquefasciatus* when compared to *Ae. aegypti*, we probably wouldn't have enough samples to collect at all time points. Additionally, FTA classic card (Whatman® card) containing Manuka honey blend® were made available on the screens of all cages, from the 7th to the 14th dpe to ensure viral detection, through saliva collection, since females had not been collected on 7th, 14th and 21st dpe, as performed for *Ae. aegypti*.

Survival and longevity analysis was also performed among *Ae. aegypti* females in relation to viral load, represented by the RNA copy number (CN), detected by RT-qPCR. A cut-off point was determined to define the groups (with the highest and lowest viral loads) based on the median CN values found for the infected females.

## Search for blood meal

To assess the influence of ZIKV exposure on the blood meal activity of *Ae. aegypti*, females were fed with blood devoid of virus at 7, 14 and 21 dpe, for 30 minutes. After each feeding event, the completely engorged females were selected and counted. For logistical reasons and a reduction in the number of females throughout the experiment, which would imply problems in statistical analysis, unfortunately it was not possible to evaluate all moments for all infections. For *Cx. quinquefasciatus* exposed to ZIKV and *Ae. aegypti* exposed to CHIKV, evaluations were carried out exclusively with the first post-exposure blood meal (7 dpe).

## RNA extraction and RT-qPCR

Females collected at 7, 14 and 21 dpe, as well as those that died during the study, were placed separately in 1.5 ml microtubes, containing 300 µl of a mosquito diluent and stored at -80°C until RNA extraction and RT-qPCR, described in Guedes et al. [30], with some modifications. The *primers* used for detection of CHIKV and ZIKV viral particles are described in Lanciotti et al. [43, 44]. Virus detection was performed by quantitative RT-qPCR on a QuantStudio® 5 Real-Time PCR system (Thermo Fisher Scientific, Waltham, MA, USA), according to conditions described in Guedes et al. [30]. Cycle threshold values (Ct) were used to estimate the amount of viral RNA, using the standard curve as a reference for each RT-qPCR assay, obtained through isolated transcripts from the ZIKV BRPE243/2015 and CHIKV PE480/2016 strains, as described in Kong et al. [45]. Negative controls for the feeding experiment and RT-qPCR consisted of mosquitoes fed with virus-free blood and water, respectively.

Whole mosquitoes were processed, except for *Ae. aegypti* collected at 7 dpe, whose abdomens and thorax were analyzed separately from the heads. To calculate the infection rate (IR), the number of positive females was divided by the total number of analyzed samples. To calculate the dissemination rates (DR), head samples or positive cards were used, divided by the total number of positive samples. Females and cards with Ct lower than 38 were considered as positive.

The FTA cards were placed in 1.5 mL tubes and stored at -80°C until use. To prepare the inoculum, cards were cut using multipurpose scissors and placed in 1.5 mL tubes. Next, 400 µL of ultrapure water was added to each tube, following homogenization for 5 times for 10 seconds, with 5 minute-intervals. Finally, the cards were transferred to a 10 mL syringe and filtered to enable recovery of the eluate only. The prepared inoculums were stored at −80°C until RNA extraction and RT-qPCR were performed, following the same protocol used for detection of viral RNA in mosquitoes.

## Statistical analysis

A descriptive analysis was performed: the variables were presented through graphs, followed by the presentation of the confidence interval and the *p*-value. Normality assumptions were made by applying the Shapiro Wilk tests. To assess the differences in means for the independent variables, the T-Student test was used, when the assumptions of normality were met. Otherwise, the Mann-Whitney test was applied, and the medians were evaluated. Also, the Bartlett test was used to assess homogeneity. When the assumption of homogeneity was met, the ANOVA mean test was used with Tukey's post hoc test; if not, the median was evaluated by applying the Kruskal-Wallis test, with the post hoc test for Fisher's test. The survival curve was determined using the Kaplan-Meier plot. The Cox test was applied to assess survival, and the proportionalities were evaluated using the Schoenfeld test. Conclusions were taken at a significance level of 5%. Results of the analysis were obtained using the R Core Team [46].

### Ethics statement

Ethical committee: This project was approved by the Research Ethics Committee of the Instituto Aggeu Magalhães-Fiocruz, Brazil (CAAE: 51012015.9.0000.5190).

## Results

### Infection and dissemination rates of Zika (ZIKV) and chikungunya (CHIKV) viruses

Infection and dissemination rates varied between the viruses and colonies evaluated, being higher in the experiments carried out with the chikungunya virus (IR of 96.00% for RecL and 93.00% for AeCamp). The dissemination rates of the same virus ranged between 100.00% and 94.00% for RecL and AeCamp, respectively (Table 1). With Zika virus, these values were lower in CqSLab (22.50% IR and 57.00% DR), in relation to the two colonies of *Ae. aegypti* (Table 1).

The percentage of positive cards varied among the colonies and viruses analyzed. In mosquitoes infected with ZIKV, the percentage rate was 30% for RecL, and 60% for AeCamp, while for CqSLab from the first experiment, the percentage was low: 14.28%. As cards from the negative control showed the presence of viral particles in two of the three experiments analyzed by RT-qPCR, it was not possible to analyze samples from the second and third experiments carried out with CqSLab. Surprisingly, no cards were found to be positive for CHIKV in the two colonies (RecL and AeCamp), and we have no explanation for this.

Females infected with ZIKV or CHIKV, from both colonies of *Ae. aegypti*, had similar viral load, comparing those who died or were collected in the periods from 8 to 14 and from 15 to 21 dpe. However, those infected with ZIKV, who died or were collected by the 7 dpe, had a lower viral load in both colonies (Table 2 and S1 Fig in S1 File). This difference was not observed among those infected with CHIKV (Table 2).

The ZIKV viral load (number of RNA copies per mL—CN) was significantly higher ($p = 0.019$) among females from the RecL colony who underwent a second blood meal in blood free of viral particles, at 7 dpe). The mean number of RNA copies increased from 6,13E+06 among those who did not have a second meal, to 8,26E+06 among those who ingested blood at 7 dpe. As a function of time of life after infection, it was found that the CN was significantly higher among RecL females that had completed engorgement at 7 dpe ($p = 0.003$) and died between the 8th and 22nd dpe (Fig 1 and S1 Table in S1 File). Although it was not statistically significant ($p = 0,075$), there was an increase in CN in the period between 8 and 22 days in AeCamp (Fig 1 and S1 Table in S1 File).

### Biological cost of infection with Zika (ZIKV) and chikungunya (CHIKV) viruses

In general, ZIKV and CHIKV had a significant impact on the parameters of vectorial capacity of the evaluated colonies. This impact negatively altered the reproductive capacity of females exposed to artificial oral infection by these arboviruses.

**Table 1. Zika (ZIKV) and chikungunya (CHIKV) virus infection (IR) and dissemination (DR) rates detected in laboratory *Aedes aegypti* (RecL), field *Aedes aegypti* (AeCamp) and laboratory *Culex quinquefasciatus* (CqSLab) colonies.**

| Virus | Colonies | IR | DR |
|---|---|:---:|:---:|
| ZIKV | RecL | 75.00 | 90.00 |
| | AeCamp | 68.00 | 70.00 |
| | CqSLab | 22.50 | 57.00 |
| CHIKV | RecL | 96.00 | 100.00 |
| | AeCamp | 93.00 | 94.00 |

**Table 2. Mean number of RNA copies (CN) detected at periods ≤7; 8 to 14 and 15 to 21 days post-exposure dpe to Zika virus or CHIKV, in colonies of *Aedes aegypti* in the field (AeCamp), in laboratory (RecL) and in laboratory colony *Culex quinquefasciatus* (CqSLab).**

| | Period (days) | N | Min | Max | Mean | Median | Standard deviation | EP | p value[1] | p value[2] | Significance |
|---|---|---|---|---|---|---|---|---|---|---|---|
| **ZIKV** | **AeCamp** | | | | | | | | | | |
| | ≤ 7 | 21 | 8.99E+06 | 1.29E+12 | 9.56E+10 | 4.94E+09 | 2.87E+11 | 6.26E+10 | **< 0.001** | | ≤ 7 * 8 to 14 |
| | 8 to 20 | 20 | 2.96E+08 | 3.12E+12 | 6.74E+11 | 2.50E+14 | 9.3E+11 | 2.08E+11 | | **0.037** | |
| | 15 to 21 | 17 | 2.89E+07 | 4.42E+12 | 3.97E+11 | 2.55E+10 | 1.05E+12 | 2.55E+11 | | | |
| | **RecL** | | | | | | | | | | |
| | ≤ 7 | 28 | 1.99E+07 | 6.55E+11 | 5.07E+10 | 4.31E+08 | 1.46E+11 | 2.75E+10 | **< 0.001** | | ≤ 7 * 8 to14 |
| | 8 to 20 | 24 | 6.40E+07 | 5.07E+12 | 5.48E+11 | 1.46E+11 | 1.12E+12 | 2.3E+11 | | **0.002** | ≤ 7 * 15 to 21 |
| | 15 to 21 | 13 | 9.00E+07 | 8.23E+12 | 1.09E+12 | 1.17E+11 | 2.37E+12 | 6.57E+11 | | | |
| **CHIKV** | **AeCamp** | | | | | | | | | | |
| | ≤ 7 | 17 | 6.52E+10 | 2.54E+12 | 1.14E+12 | 8.98E+11 | 8.75E+11 | 2.12E+11 | **0,030** | 0,806 | |
| | 8 to 20 | 19 | 4.61E+08 | 4.71E+12 | 1.42E+12 | 1.26E+12 | 1.54E+12 | 3.54E+11 | | | |
| | 15 to 21 | 17 | 4.00E+09 | 5.88E+12 | 1.29E+12 | 5.82E+11 | 1.73E+12 | 4.18E+11 | | | |
| | **RecL** | | | | | | | | | | |
| | ≤ 7 | 23 | 1.E+07 | 8.09E+12 | 1.53E+12 | 1.61E+11 | 2.45E+12 | 5.12E+11 | **0,048** | 0,880 | |
| | 8 to 20 | 17 | 1.54E+10 | 7.5E+12 | 1.63E+12 | 3.58E+11 | 2.31E+12 | 5.61E+11 | | | |
| | 15 to 21 | 16 | 1.16E+10 | 4.33E+12 | 9.94E+11 | 3.99E+14 | 1.33E+12 | 3.32E+11 | | | |

## Survival and longevity

The analysis of the survival curve of the groups exposed to ZIKV, showed that the risk of death for females from RecL was about twice as high (E = 1.845: p = 0.014 and EI = 2.014: p = 0.003), compared to the control group (non-exposed—NE) (Fig 2A and S2 Table in S1 File). Mean lifespans for the three groups did not differ significantly: 33.36; 31.04 and 26.23 days for NE, E and EI, respectively. However, for AeCamp, there was no significant difference in survival between the three groups analyzed, as the risk of death was 1,289 for group E and 1,212 for EI (Fig 2B and S2 Table in S1 File). The average lifespan in AeCamp was also similar for the three groups: 34.19; 31.19 and 31.26 days, for NE, E and EI, respectively. Among *Cx. quinquefasciatus* females, survival (E = 1.300 and EI = 0.805) (Fig 2C and S3 Table in S1 File). and longevity (34, 73; 31, 34 and 39, 50 days, for NE, E and EI, respectively) were not altered among females exposed to ZIKV.

Likewise, the results found for the RecL colony, after exposure to CHIKV, showed an impact of infection on survival, with a higher risk of death (3.963) for the EI group (Fig 3A and S4 Table in S1 File); however, this difference appeared only between day zero and the 20th day of observation (p = 0.001), i.e., it was not found after the 21st day (Fig 3B). The longevity of RecL females was also reduced by CHIKV infection: 38 and 17 days for NE and E, respectively (p = 0.002). On the other hand, in AeCamp, the survival curves showed no statistical difference between the two study groups (Fig 3C and S1-S5 Tables in S1 File). Longevity was 37.5 days for the control group (non-exposed–NE) and 32 days for the EI group.

In general, there was no correlation between longevity and viral load. The only exception was observed for the longevity of CHIKV infected AeCamp females, where there was a correlation identified as negative (the greater the longevity, the lower the viral load detected, - 0.534, p = 0.033). It should be noted that, from this analysis, females collected while still alive were removed from analysis (S6 Table in S1 File).

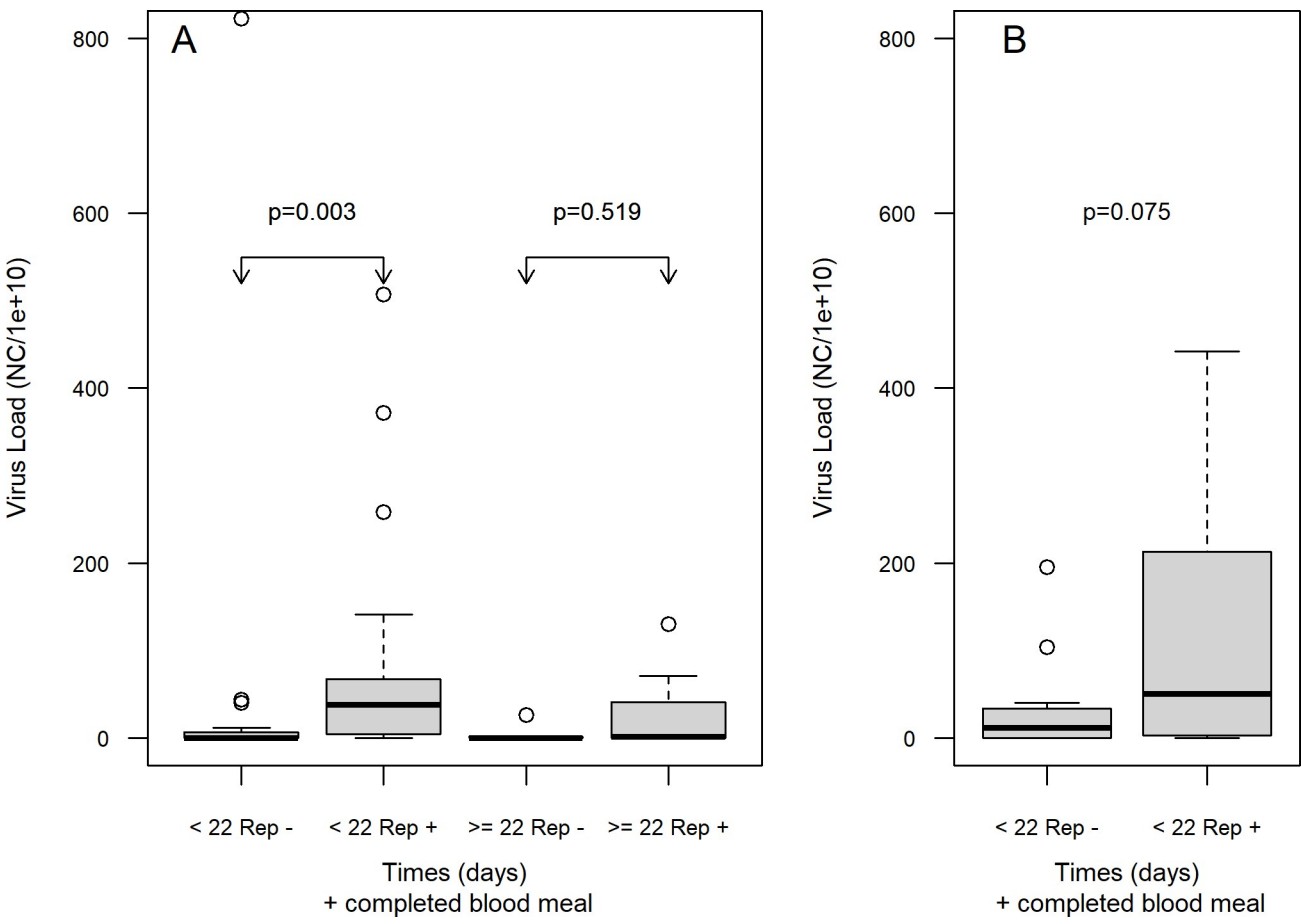

**Fig 1. Viral load (number of RNA copies of ZIKV–CN, per mL, detected by RT-qPCR), in *Aedes aegypti* colonies from the laboratory—RecL and field—AeCamp, infected with Zika virus (ZIKV), which completed blood meal on the 7th day post-exposure (dpe), in blood free of viral particles, as a function of post-exposure life span to the virus.** A: RecL colony; B: AeCamp colony. Notes: < 22 Rep—: females that did not have a blood meal at 7 dpe post-exposure to ZIKV and died until the 21st dpe; > = 22 Rep—: females that did not have a blood meal at 7 dpe post-exposure to ZIKV and died after the 21st dpe; < 22 Rep +: females who blood fed at 7 dpe to ZIKV and died until the 21 st dpe; > = 22 Rep +: females that had a blood meal at 7 dpe post-exposure to ZIKV and died after the 21st dpe.

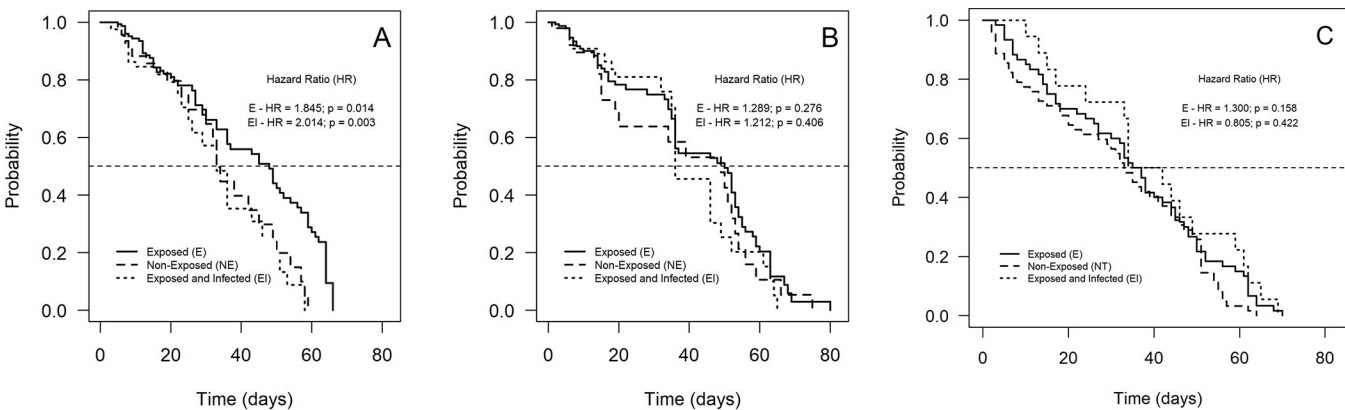

**Fig 2.** Survival curve of *Aedes aegypti* females–laboratory colony–RecL, over 66 days of observation; field colony—AeCamp, over 80 days of observation, and laboratory colony of *Culex quinquefasciatus*—CqSLab, over 70 days of observation, after exposure to Zika virus (ZIKV). A–RecL colony; B–AeCamp colony; C–CqSLab colony.

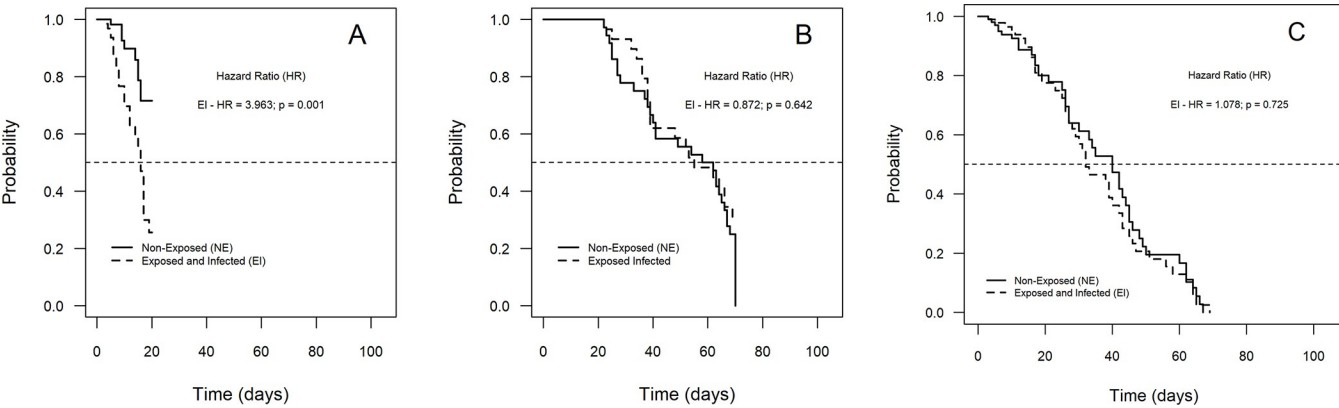

**Fig 3. Survival curves: individuals from *Ae. aegypti* from laboratory—RecL, over 70 days and field—AeCamp, over 69 days of observation, after exposure to chikungunya virus (CHIKV).** A—RecL Colony—observation period less than or equal to 20 days; B—RecL Colony—observation period longer than 20 days; C—AeCamp Colony–observation period of 69 days.

## Fecundity and fertility in the first gonotrophic cycle

Fecundity (average number of eggs) and fertility (average percentage of larvae hatching) were not altered by exposure to ZIKV among *Ae. Aegypti* from the RecL colony (Fig 4A and S7 Table in S1 File, in the first gonotrophic cycle (number of eggs in the NE group = 87.83; E = 79.13; EI = 82.07 and average hatching percentage NE = 67.36%; E = 66.31%; EI = 69.12%). In AeCamp, fecundity was also unaltered (number of eggs NE = 72.70; E: 70.40 and EI: 70.90) by exposure to ZIKV, although a significant reduction occurred in fertility (average percentage of hatching NE = 65.46%; E: 50.68% and EI: 49.38%) (Fig 4B and S7 Table in S1 File). In *Cx. quinquefasciatus*, these parameters were not altered among infected females. However, fecundity has been reduced in those exposed to the virus and which did not develop the infection (number of eggs = NE: 102.25; E: 76.90; EI: 92.67. (Fig 4C and S7 Table in S1 File).

The results show that the fecundity of females from the RecL colony was not significantly altered by the infection with CHIKV in the first gonotrophic cycle. However, the infection impacted fertility, with a reduction in the median percentage of hatching from 63.48% in group NE to 40.67% in group EI (Fig 5 and S8 Table in S1 File). Differently, in AeCamp, CHIKV had an impact on fecundity, reducing the median number of eggs from 48 in NE to 38 in group EI. The fertility of this colony was also altered by the infection, with a reduction in the median percentage from 57.50% to 37.50%, between groups NE and EI, respectively (Fig 5 and S8 Table in S1 File).

As with longevity, the correlation between reproductive capacity variables (fecundity and fertility) and viral load (CN) was analyzed. For CHIKV-infected RecL, the correlation was positive between CN and fecundity (0.332, p = 0.011) and negative between CN and fertility (-0.388, p = 0.003). In AeCamp, infected with the same virus, there was no correlation between these parameters, as well as for all colonies infected with ZIKV (S6 Table in S1 File).

## Blood feeding activity

The blood meal activity of RecL, AeCamp and CqSLab colonies was not altered by exposure to ZIKV and CHIKV, regarding the search for a second blood meal. Detailed numbers are shown in Table 3.

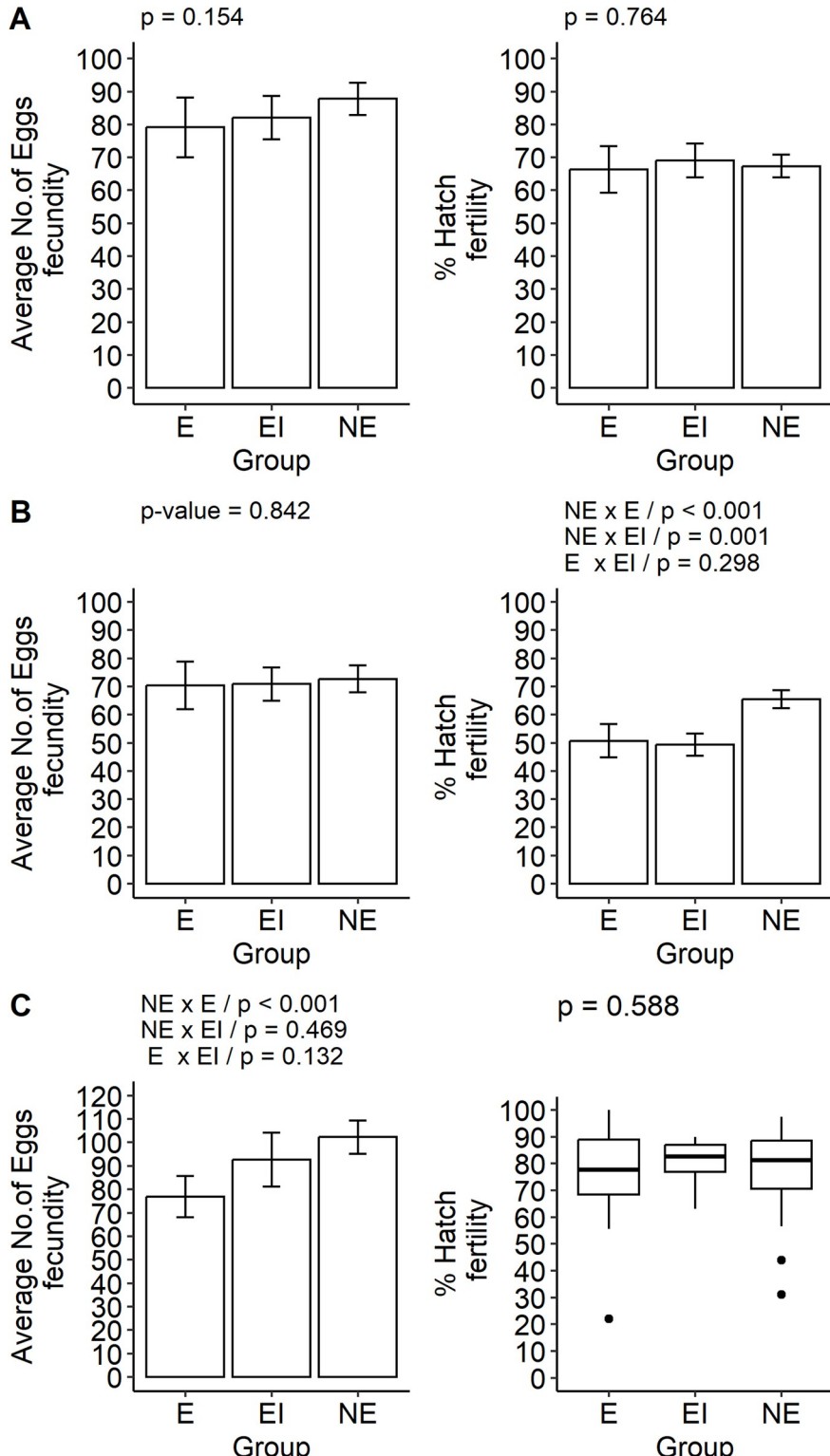

**Fig 4. Average number of eggs laid and average percentage of larval hatching in the first gonotrophic cycle of** *Aedes aegypti* **females–laboratory colony–RecL and field colony–AeCamp, and of** *Culex quinquefasciatus* **females–laboratory colony–CqSLab, after exposure to Zika virus (ZIKV).** Note: NE–Control (non-exposed); EI—exposed and infected. A: RecL Colony; B: AeCamp Colony; C: CqSLab Colony.

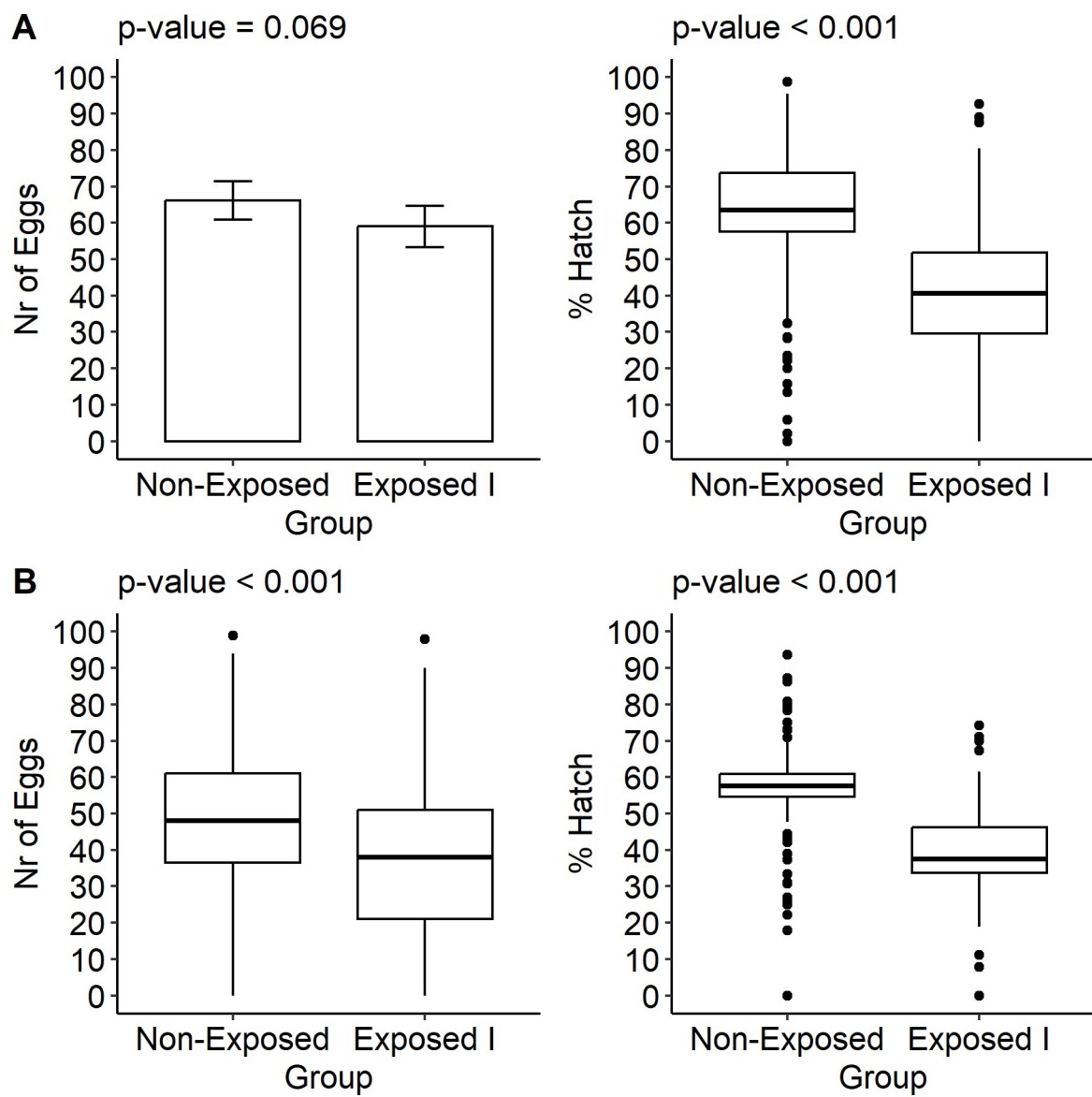

**Fig 5. Mean or median number of eggs and median percentage of hatching of *Aedes aegypti* larvae–laboratory colony—RecL and field colony—AeCamp, in the first gonotrophic cycle, after exposure to chikungunya virus (CHIKV).** Note: NE–Control (non-exposed); EI—exposed and infected. A: RecL Colony; B: AeCamp Colony.

## Discussion

Relevant aspects of the biological performance of vector mosquitoes can be altered by the process of pathogen infection, e.g., parameters involved in vectorial capacity. This event may lead to a consequent change in the pattern of occurrence of an epidemic in a given epidemiological context [1]. Studies have shown the biological cost of arbovirus infection, namely reduced survival, longevity, reproductive capacity, blood feeding, among other impacts in mosquitoes. The different results found between them are explained by the direct relationship between the virus lineages and vectors involved [3, 5–8, 47]. Significant changes in reproductive capacity, for example, can limit the number of offspring of infected females and determine the

**Table 3. Number and percentage of females of *Aedes aegypti*, from colonies RecL and AeCamp, and of *Culex quinquefasciatus*, from colony CqSLab, which completed the blood meal in weeks following exposure to Zika virus (ZIKV) and from colonies RecL and AeCamp exposed to chikungunya (CHIKV), in blood free of viral particles.**

| Virus | Colonies | dpe | Groups | Blood Meals | | | | | *p*-value |
|---|---|---|---|---|---|---|---|---|---|
| | | | | Negative (-) | | Positive (+) | | | |
| | | | | No. | % | No. | % | Total number | |
| ZIKV | RecL | 7 | Non-exposed | 42 | 35.59 | 76 | 64.41 | 118 | 0.695 |
| | | | Exposed | 16 | 43.24 | 21 | 56.76 | 37 | |
| | | | Exposed and infected | 22 | 38.6 | 35 | 61.4 | 57 | |
| | | 14 | Non-exposed | 45 | 55.56 | 36 | 44.44 | 81 | 0.526 |
| | | | Exposed | 15 | 51.72 | 14 | 48.28 | 29 | |
| | | | Exposed and infected | 14 | 43.75 | 18 | 56.25 | 32 | |
| | | 21 | Non-exposed | 21 | 72.41 | 8 | 27.59 | 29 | |
| | | | Exposed | 7 | 43.75 | 9 | 56.25 | 16 | 0.085 |
| | | | Exposed and infected | 14 | 77.78 | 4 | 22.22 | 18 | |
| | AeCamp | 7 | Non-exposed | 61 | 52.59 | 55 | 47.41 | 116 | 0.142 |
| | | | Exposed | 24 | 64.86 | 13 | 35.14 | 37 | |
| | | | Exposed and infected | 30 | 57.69 | 22 | 42.31 | 52 | |
| | | 14 | Non-exposed | 56 | 70 | 24 | 30 | 80 | 0.342 |
| | | | Exposed | 31 | 62 | 19 | 38 | 50 | |
| | | | Exposed and infected | 18 | 56.25 | 14 | 43.75 | 32 | |
| | CqSLab | 7 | Non-exposed | 19 | 34.55 | 36 | 65.45 | 55 | 0.771 |
| | | | Exposed | 15 | 28.85 | 37 | 71.15 | 52 | |
| | | | Exposed infected | 5 | 27.78 | 13 | 72.22 | 18 | |
| CHIKV | RecL | 7 | Non-exposed | 46 | 57.50 | 34 | 42.50 | 80 | 1,000 |
| | | | Exposed infected | 43 | 56.58 | 33 | 43.42 | 76 | |

transmission dynamics of an arbovirus. In general, this dynamic also occurs as a function of the age at which the vector acquires the infection, as well as survival and longevity [6, 15].

In this sense, the two species investigated here, suffered a negative impact on biological performance, especially on reproductive capacity, after exposure to Zika (ZIKV) or chikungunya viruses (CHIKV), with a reduction in the number of individuals for the subsequent generation. This has evolutionary implications, once any trait that may offer an adaptive advantage for the mosquito's defense against viral infection is unlikely to be selected for. However, the ability of a vector to transmit a pathogen is multifactorial and, therefore, isolated assessments in any of its parameters must be made with caution [1].

Our results showed that the natural population (AeCamp) did not suffer any impact on its survival or longevity caused by the infection of both viruses. This may suggest that infected mosquitoes can keep transmitting the virus for long periods in the environment. The interaction between vectors and viruses, genetically determined and triggered by the adaptive process in an environment, can result in more efficient transmission dynamics, with less impact on the biological performance of *Ae. aegypti* [47].

On the other hand, for the *Ae. aegypti* colony, RecL, there was a significant reduction in survival when it was exposed and infected with both viruses. This result demonstrates a greater fragility of this colony, since the infection and dissemination rates for the two viruses were always higher when compared to the field population. It is known that the colonization process causes genetic diversity loss, which may impact on mosquito defense, metabolism, development, among other important traits for mosquito survival. Therefore, the results found in studies using laboratory colonies should always be interpreted with caution, before being

extrapolated to what actually occurs in nature. The reduction in the survival rate and number of eggs laid by *Ae. aegypti* was associated with the process of adaptation to CHIKV infection, in a study developed by Sirisena, Kumar and Sunil [6]. The authors demonstrated that there is a negative regulation of genes involved in the egg laying pathway in infected females, through the analysis of transcript expression [6].

Previous studies have addressed cost on longevity [4] and survival [47] of *Ae. aegypti* populations infected by ZIKV. Petersen et al. [4], for example, used the ZIKV strain BRPE243/2015, the same strain used in the present study, to assess the reproductive capacity and longevity of *Ae. aegypti* collected in Rio de Janeiro, and they found a negative impact of the infection on the longevity of females. Thus, owing to genetic background and environmental factors, the viral strain used for infection in the laboratory should be considered, because especially in *Ae. aegypti*, the interaction between pathogen and vector can vary even between geographically close populations [47].

Similarly, the longevity of *Ae. aegypti* from Palm Beach County, Florida was also unaltered using the CHIKV strain LR2006-OPY1 [4]. However, the same study suggests that physiological restrictions on the evolution of CHIKV infection in *Ae. albopictus* result in biological cost on this species, considering that they found a significant reduction in longevity. Additionally, the body titer of female *Ae. albopictus* infected with CHIKV and longevity were inversely correlated. For *Ae. aegypti*, the authors reported no correlation between these parameters [48]. Considering that the viral load can impact the biological performance of infected mosquitoes, the correlation between the lifetime of infected females (group EI) by ZIKV or CHIKV and the number of RNA copies (CN) in all colonies was analyzed in this study. Differently from what was found by Reiskind et al. [48] for field *Ae. aegypti*, here there was a negative correlation between longevity and the number of RNA copies/mL–CN, for females of *Ae. aegypti* from the field population infected with CHIKV. These results suggest that a high viral load impact longevity, however, it should also be considered that the virus may reduce its replication, as a consequence of the functioning of the immune system throughout the mosquito's life [48] which precludes a clear interpretation of these results. Studies report acute arbovirus replication in the first two days after infection, as well as a slow reduction in the virus body titer and elimination throughout the mosquito's lifetime. This dynamic can vary between vector species and viruses, as well as between different vector tissues [49, 50].

The number of eggs laid in the first gonotrophic cycle was not altered by exposure and infection with ZIKV in the two colonies of *Ae. aegypti*. On the other hand, the fertility of the field colony, AeCamp, was significantly reduced even among females exposed to ZIKV. According to Li et al. [51], the ovaries of mosquitoes are affected by ZIKV on the second day after infection. Despite this fact, ZIKV did not reduce the fecundity of *Ae. aegypti*, as also found by Padilha et al. [3] and Resck et al. [5], for laboratory strain, and Silveira et al. [47], for field population sample (F1). In contrast, a field population in Rio de Janeiro showed a reduction in fecundity. Interestingly, in the same study, one of the groups infected by ZIKV showed an increase in this parameter, when compered the first and the third gonotrophic cycles [4].

The negative impact on AeCamp fertility suggested that the cost resulting from exposure to the virus was directed to the viability of the eggs, regardless of the establishment of the infection. Resck et al. [5] did not find an impact of ZIKV infection on the fertility of females from an *Ae. aegypti* from laboratory, corroborating the results reported in this paper for RecL. For the other hand, Ciota et al. [52] describe the adaptive process is a determining factor for viral replication, in the case of successful infection, with subsequent increase in the immune response and consequent impact on the biological performance of the mosquito vector. The same author tested a strain of *Cx. pipiens* adapted, in the laboratory, to the West Nile Virus and observed a reduction in fecundity in the second week of the experiment, after exposure to

the virus, which did not occur with the non-adapted strain of the same species. This hypothesis can explain the results regarding the fertility of AeCamp obtained here, which had contact with the same strain used in these experiments, during the Zika fever epidemic in 2015.

Parameters of reproductive capacity, especially fertility, showed a biological cost for infection with CHIKV, for both colonies, from the laboratory or field. Resck et al. [5] evaluated the reproductive capacity of *Ae. aegypti* after infection with CHIKV and found a negative effect on fertility, but not on fecundity to laboratory colonie. A similar result was found with the CHIKV strain and *Ae. albopictus* field population from the Italy. This alteration in fertility suggests that CHIKV can affect embryonic development and embryo survival [34] as described for other viruses [4, 7]. This may also explain the negative correlation found in this study between the viral load and the fertility of the laboratory *Aedes aegypti* colony. The relevant effect of CHIKV and ZIKV infection on fertility, found in this study, suggests the need for further investigation, considering the variable time of interaction in the field between these arboviruses and the field populations, assuming that this as this seems to be a critical point in the relationship vector/parasite.

For *Cx. quinquefasciatus* (CqSLab), survival and longevity were not affected by ZIKV exposure or infection. Styer, Meola and Kramer [7] found no difference in this aspect of vectorial capacity between *Cx. tarsalis* females, in a laboratory colony, after exposure to West Nile virus (WNV). The survival of *Cx. tarsalis*, however, was altered by Western equine encephalitis virus (WEEV) infection [53]. In this study, exposure to ZIKV, but not infection, impacted the reproductive capacity of females, significantly reducing fecundity. For a laboratory colony of *Cx. pipiens*, pre-selected by continuous exposure to a WNV strain, fecundity was also altered; however, unlike what was found in the present study, infection, not just exposure, reduced this parameter [52]. The fecundity and fertility of *Cx. tarsalis* infected by WNV was also reduced in the infected groups [7]. Additionally, the same authors reported that the percentage of larvae hatching was higher in the exposed group (65.8%), than in the non-exposed (55.6%) and exposed and infected groups (42.5%).

The results analyzed for post-exposure blood feeding activity in an artificial feeder suggest that there is no effect of exposure or infection on the search for subsequent blood feeding in *Ae. aegypti* and *Cx. quinquefasciatus*, for both viruses and colonies studied here, considering the criterion evaluated (percentage of females that completed the blood meal within 30 minutes of exposure to the feeder). On the other hand, when evaluating the time spent by females of *Ae. aegypti* to complete engorgement, Sylvestre, Gandini and Maciel-de-Freitas [8] found that infected females spent more time compared to the non-exposed group. Additionally, a higher percentage of females of *Cx. tarsalis* infected with WNV had a blood meal on artificial feedings after infection, compared with unexposed females and exposed females that had not been infected. However, the same authors reported no significant difference in the amount of blood ingested between the three groups evaluated [7].

In general, the findings suggest that exposure to ZIKV and CHIKV significantly impact the reproductive capacity and longevity of the colonies evaluated. CHIKV had a greater impact on *Ae. aegypti*, in comparison to ZIKV, considering both parameters of the reproductive capacity of the field mosquitoes as well as the fertility of RecL. The greater impact observed in CHIKV infection compared to ZIKV can be explained by the different dynamics of viral dissemination, in agreement with Resck et al. [5], or by the shorter extrinsic incubation period (around two days) observed for this virus by Fuller et al. [54], although ZIKV was detected in the ovaries and salivary glands as early as the second day after infection [51]. The fact that ZIKV did not alter the fecundity of *Ae. aegypti*, in our study, may be related to the low viral detection in ovaries, found by Nag et al. [55] on the tenth day after infection. The same authors reported an increased prevalence of ovarian infection after a second blood meal in blood free of viral

particles [55]. It is noteworthy that, in this study, only the data from the first gonadotrophic cycle were statistically analyzed, since a robust analysis was not possible for the data collected in the second and third cycles (data not shown).

Only exposure to ZIKV, but not infection, was enough to reduce the fecundity of *Cx. quinquefasciatus* females, suggesting that the triggering of defense mechanisms associated with the midgut barriers generates a biological cost for the species. Although *Cx. quinquefasciatus* showed much lower infection and dissemination rates than *Ae. aegypti*, this species is much more abundant in the environment in Recife; thus, its role in ZIKV transmission is not clear yet. In *Ae. aegypti*, ZIKV infection reduced the fertility of field females, but not their fecundity. However, this impact may have little relevance, considering that longevity, survival and search for blood source after exposure to the virus were not affected in the field population.

In the colony of *Ae. aegypti*, females infected with ZIKV, which had a second blood meal at 7 dpe, had a significantly higher number of RNA copies, compared to those that did not have a second meal. Similarly, Cui et al. [56] reported a three to four-fold increase in viral load among *Ae. aegypti* infected with DENV-4 after blood feeding at 5 dpe. However, owing to the limited number of females analyzed for this aspect, new experiments should be carried out, as they may allow a better statistical evaluation of this relationship, involving other variables to possibly confirm this result.

In summary, the results presented here confirm the hypothesis that there is a negative impact of infection by the Zika and chikungunya viruses on the reproductive capacity parameters evaluated. However, this hypothesis has not been confirmed regarding the impact on longevity and hematophagous behavior. It is worth mentioning that a limitation of this study was the analysis of only samples from the first gonotrophic cycle which means that other effects of ZIKV and CHIKV infections in mosquitoes, that were not possible to be observed here, could be observed in later cycles, as pointed out by other authors [55, 57]. It is suggested, in the case of *Ae. aegypti* field population, that there was an adaptive process that favored these parameters, probably resulting from the previous contact of this mosquito population, during the ZIKV and CHIKV epidemics, between 2015 and 2016, with the virus strains studied here. In *Cx. quinquefasciatus*, the results showed lower susceptibility to ZIKV, which may justify the lower impact on this colony, as reported by Reiskind et al. [48].

Additionally, our results corroborate the hypothesis of *Cx. quinquefasciatus* participation in the ZIKV transmission process [30]. Finally, we highlight the relevance and novelty of this study, especially considering the lack of laboratory investigations to address aspects of blood-feeding behavior, after exposure to ZIKV and CHIKV, in *Ae. aegypti* and *Cx. quinquefasciatus*.

## Supporting information

**S1 File.**
(DOCX)

## Acknowledgments

The authors would like to thank the Reference Service for Control of Culicidae Vectors and the insectary team for their support and supply of mosquitoes. To the Secretary of Health of Recife. Larissa Krokovsky for help with RT-qPCR.

## Author Contributions

**Conceptualization:** Mônica Crespo, Duschinka Guedes, Marcelo Paiva, George Tadeu, Claudia Oliveira, Rosângela Barbosa, Constância Ayres.

**Data curation:** Mônica Crespo, George Tadeu.

**Formal analysis:** Mônica Crespo, George Tadeu.

**Funding acquisition:** Marcelo Paiva, Constância Ayres.

**Investigation:** Constância Ayres.

**Methodology:** Mônica Crespo, Duschinka Guedes, Mariana Sobral, Elisama Helvecio, Rafael Alves.

**Writing – original draft:** Mônica Crespo.

**Writing – review & editing:** Marcelo Paiva, Claudia Oliveira, Maria Alice Varjal Melo-Santos, Rosângela Barbosa, Constância Ayres.

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
