## [Decision Letter · Decision Letter 0]

18 May 2023

PONE-D-23-02967Exposure to Zika and chikungunya viruses impacts aspects of the vectorial capacity of Aedes aegypti and Culex quinquefasciatusPLOS ONE

Dear Dr. Guedes,

Thank you for submitting your manuscript to PLOS ONE. After careful consideration, we feel that it has merit but does not fully meet PLOS ONE’s publication criteria as it currently stands. Therefore, we invite you to submit a revised version of the manuscript that addresses the points raised during the review process.

We look forward to receiving your revised manuscript.

Kind regards,

Kelli L. Barr, Ph.D.

Academic Editor

PLOS ONE

Journal Requirements:

3. Thank you for stating the following in your Competing Interests section:  "The authors declare no conflict of interest"

4. Please amend the manuscript submission data (via Edit Submission) to include author: Constância Ayres

**Additional Editor Comments:**

Two of the reviewers raised significant concerns about the results with reviewer 3 raising the most important issues. It looks like you are comparing 2 different populations of mosquitoes that have been treated differently. This must be clarified.

Reviewers' comments:

Reviewer's Responses to Questions

**Comments to the Author**

1. Is the manuscript technically sound, and do the data support the conclusions?

Reviewer #1: Yes

Reviewer #2: Partly

Reviewer #3: No

2. Has the statistical analysis been performed appropriately and rigorously? 

Reviewer #1: Yes

Reviewer #2: Yes

Reviewer #3: No

3. Have the authors made all data underlying the findings in their manuscript fully available?

Reviewer #1: No

Reviewer #2: Yes

Reviewer #3: Yes

4. Is the manuscript presented in an intelligible fashion and written in standard English?

Reviewer #1: Yes

Reviewer #2: Yes

Reviewer #3: No

5. Review Comments to the Author

Reviewer #1: In the following manuscript by Crespo and colleagues entitled “Exposure to Zika and chikungunya viruses impacts aspects of the vectorial capacity of Aedes aegypti and Culex quinquefasciatus”, the authors examined the effect of CHIKV and ZIKV infection on the vectorial capacity of natural and laboratory stablished colonies of Ae. aegypti and Culex quinquefasciatus. Their results found that CHIKV and ZIKV altered some biological parameters in Ae. aegypti and Culex quinquefasciatus including fecundity, fertility, survival, longevity, and blood feeding status. These results provide a better understanding over the processes of virus-vector interaction and can shed light on the complexity of arbovirus transmission.

Elucidating the mechanisms of viral and vector interactions that determine the vector competence of insect species such as mosquitoes is critical to understand the dynamics of transmission of these viruses among vector populations and their transmission to human hosts. This manuscript by Crespo and colleagues addressed some biological aspects of the arbovirus infections of highly anthropogenic and competent mosquito species, Aedes aegypti and Culex quinquefasciatus from an area with high transmission for ZIKV and CHIKV such as the case of Brazil. The manuscript is well-written and the results well-described; however, these are still some aspects that needed to be clarified by the authors as noticed below:

Major(s)

1. Knowing that the vertical transmission of arboviruses such as dengue virus (DENV), ZIKV and CHIKV may occur in endemic areas; were natural isolated colonies of Ae. aegypti previously tested for DENV infection? This is important to know as pre-established viral infections in mosquito populations may affect their vectorial capacity once evaluated under laboratory conditions? RT-PCR? Was any other marker of arboviral infection is mosquitoes used?

2. Knowing the maturation stage that occur in flavivirus virions such as ZIKV and DENV once grown in either mammalian (e.g., Vero) or insect cells (C6/36 cells), was any experiment performed using viruses derived from infected mosquito cells?

3. For viral infection of both mosquito species, the authors used as negative control a mixture of equal volume of virus free cell culture and defibrinated rabbit blood; however, knowing that viral replication after infection of mosquitos would be the main cause of the altered biological parameters in mosquitoes, would not be better using inactivated viruses during blood feeding experiments instead?

4. In the methods section the authors stated that “Whole mosquitoes were processed, except for Ae. aegypti collected at 7 dpe, whose abdomens and thorax were analyzed separately from the heads. How is that these important data was not included at least as a supporting information in this manuscript? This is a critical biological parameter for measuring vector competence and the level of viral infection (e.g., viral RNA load, viral genome equivalents, etc) detected in each individual mosquito that may affect the conclusions described in the manuscript. Please clarify this and add this set of data to the manuscript.

Reviewer #2: Peer Review for:

Manuscript Number: PONE-D-23-02967

Manuscript Title: Exposure to Zika and chikungunya viruses impacts aspects of the vectorial capacity of Aedes aegypti and Culex quinquefasciatus

In their manuscript Crespo et al explore the impact of ZIKV and CHIKV arbovirus exposure and infection on reproductive capacity, longevity, and blood feeding of Aedes aegypti and Culex quinquefasciatus mosquitoes. They accomplish this by exposing lab and field-caught colonies of Ae. aegypti to both ZIKV and CHIKV, and a lab colony of Cx. quinquefasciatus to ZIKV and then assessing fecundity and fertility. To evaluate survival and longevity, the authors collected Ae. aegypti mosquitoes at 7, 14, and 21 days post exposure, but only collected Cx. quinquefasciatus when death occurred throughout the experiment. FTA cards infused with Manuka honey blend were utilized for viral detection. Additionally, blood-seeking behavior was assessed at 7, 14, and 21 days post exposure. RT-qPCR was performed for viral detection of the samples. While it’s significant that both viruses were associated with a biological cost, in particular, diminished reproductive capability, the fact that blood feeding activity remained the same (or even increased) after exposure helps further explain viral dissemination and global expansion of these arboviruses.

Major Issues:

Line 170-171 - Why were Cx. quinquefasciatus mosquitoes treated differently (evaluated differently) from Ae. aegypti?

Line 244-247 - More information is needed concerning the contamination of samples and inability to carry out 2nd and 3rd experiments. Did you consider re-running the experiment for the sake of replication?

Figures 1-3 are not visible on the downloaded PDF.

Minor Issues:

Please see my edited version of your manuscript with highlights and comments.

Line 28 – no period is needed after “AeCamp”

Line 38-39 – consider defining “vectorial capacity” a bit here. You may also consider differentiating “vectorial capacity” from “vector competence”

Line 41-43 – The cited study involves Anopheles mosquitoes and doesn’t relate well as an “example” of what you previously stated.

Line 67 – “vector” should read “vectorial”

Line 68-70 – consider re-wording this sentence as it is hard to understand

Line 71 – use commas instead of semicolons.

Line 83-84 – This could use a reference.

Line 100 – What is “shelling”?

Line 113 – “tittered” should be “titered”

Line 181 – Consider including the timeframe that mosquitoes were allowed to seek a blood meal.

For Figure captions, please include a clear description of what group each panel represents. For example, you do this in Fig 5 caption (A: RecL Colony; B: AeCamp Colony), but not clearly for Fig 4.

Line 359 – Consider “blood feeding” instead of “blood meal activity”

Line 377 – remove the “a”.

Line 494 – Consider changing “of control of” to “to control”

Overall, this is a well designed and executed study to further explore the biological cost of arbovirus exposure and the impact of infection on vectorial capacity. Recommend acceptance for publication after minor revisions.

Reviewer #3: The paper discusses such an important issue, the effects of arbovirus on the biology of two insect vectors, Aedes aegypti, and Culex quinquefasciatus. Despite the interesting question, some details should be clarified.

It was challenging to analyze the results as the materials and methods needed to be clarified. It seems the authors made lots of exceptions in the experimental designs. If it is not a description problem, the results are not comparable because they were not made the same way.

Introduction

Lines 41 – 43: “For example, the number of eggs produced in the first gonotrophic cycle indicates the total lay profile during the entire female life in mosquitoes [10].” I think some more recent references do not confirm this statement. Can you check this information?

Materials and methods

Lines 116 – 117: “For Ae. aegypti, each experiment was performed with two groups for each colony (RecL, 117 AeCamp, and CqSLab)”. CqSLab is a colony of C. quinquesfasciatus, isn’t it? I presume you meant one of the viruses, right? This point needs to be clarified.

Lines 133 – 134: How did you group the exposed but not infected (E) samples for ZIKV? Did you check all the mosquitoes individually to split them into the E and EI groups? This point needs to be clarified.

Lines 136 – 140: Did you discard the E group for CHIKV or join them with the EI group?

Lines 170 – 174: Survival and longevity assessment in Culex – why didn’t you repeat the same experimental design as Aedes? Why didn’t you collect samples at 7, 14, and 21 DPE?

Lines 182 – 184: “For Cx.quinquefasciatus exposed to ZIKV and Ae. aegypti exposed to CHIKV, evaluations were carried out exclusively with the first post-exposure blood meal (7 dpe).”. Again, a variation in the experimental design happens here. Why didn’t you perform the experiments in the same way? It should be stated in the M&M section if it was a methodological problem.

Lines 185-210: RNA extraction and qPCR assays: a) I need clarification on whether you tested all the females or only the females collected at 7, 14, and 21 dpe. b) “To calculate the infection rate (IR), the number of positive females was divided by the total number of analyzed samples” What is the size of the samples? c) regarding the section ‘Assessment of the biological parameters after exposure to viruses,’ did you calculate the IR based on subsamples?

The viral detection of females needs a more precise explanation.

Lines 223-225: This reference should be in the References Section.

Results

Lines 258-267: what is the total number of females analyzed in each group?

Lines 283 – 286: I didn’t find the data related to Cx. quinquefasciatus females infected females in the Supplementary tables.

(S1 Table. Risk of death for females of Ae. aegypti, RecL and AeCamp colonies, after exposure to Zika virus (ZIKV); S2 Table. Risk of death for females from the laboratory Aedes aegypti colony – RecL after exposure to chikungunya virus (CHIKV) during the first 20 days of observation; S3 Table. Risk of death for females from the field colony of Aedes aegypti – AeCamp after exposure to chikungunya virus (CHIKV).)

Lines 288 – 291: The line format used in the graphs is very confusing. Can you change them to avoid confusion?

Discussion

In general, the discussion section is superficial. The data presented sometimes supports and sometimes disagrees with the data in the literature. Still, it needs to be clarified what is the actual contribution of the findings to the vector-host interaction area.

6. PLOS authors have the option to publish the peer review history of their article (what does this mean?). If published, this will include your full peer review and any attached files.

Reviewer #1: **Yes: **Henry Puerta-Guardo

Reviewer #2: **Yes: **Mark F. Olson

Reviewer #3: No

---

## [Author Response · Author response to Decision Letter 0]

9 Oct 2023

Dear Kelli L. Barr, 

Academic Editor PLOS ONE

We thank you and three reviewers for improving our manuscript. Here is our answer to each question raised during manuscript review.

Journal Requirements:

R: Thanks, we have done that

2. We note that the grant information you provided in the ‘Funding Information’ and ‘Financial Disclosure’ sections do not match

R: We have corrected that

3. Thank you for stating the following in your Competing Interests section: "The authors declare no conflict of interest"

R: We have done that

4. Please amend the manuscript submission data (via Edit Submission) to include author: Constância Ayres

R: My name is already there. I am the last author.

R: We have corrected that

 Additional Editor Comments:

Two of the reviewers raised significant concerns about the results with reviewer 3 raising the most important issues. It looks like you are comparing 2 different populations of mosquitoes that have been treated differently. This must be clarified.

R: We have answered these issues

Please do not edit Reviewers' comments:

Reviewer's Responses to Questions

Comments to the Author

1. Is the manuscript technically sound, and do the data support the conclusions?

Reviewer #1: Yes

Reviewer #2: Partly

Reviewer #3: No

R: We have changed the conclusion paragraph at the end of the discussion section

2. Has the statistical analysis been performed appropriately and rigorously?

Reviewer #1: Yes

Reviewer #2: Yes

Reviewer #3: No

R: We have performed more statistical analysis and added to the manuscript

3. Have the authors made all data underlying the findings in their manuscript fully available?

Reviewer #1: No

Reviewer #2: Yes

Reviewer #3: Yes

R: We have added more data in Supplementary material

Author’s feedback: tables with data on the number of samples, standard deviation, means, medians and variance measures are now available in the supplementary material. The citations of the referred tables were inserted throughout the manuscript.

4. Is the manuscript presented in an intelligible fashion and written in standard English?

Reviewer #1: Yes

Reviewer #2: Yes

Reviewer #3: No

R: We have corrected minor errors

5. Review Comments to the Author

Reviewer #1: In the following manuscript by Crespo and colleagues entitled “Exposure to Zika and chikungunya viruses impacts aspects of the vectorial capacity of Aedes aegypti and Culex quinquefasciatus”, the authors examined the effect of CHIKV and ZIKV infection on the vectorial capacity of natural and laboratory stablished colonies of Ae. aegypti and Culex quinquefasciatus. Their results found that CHIKV and ZIKV altered some biological parameters in Ae. aegypti and Culex quinquefasciatus including fecundity, fertility, survival, longevity, and blood feeding status. These results provide a better understanding over the processes of virus-vector interaction and can shed light on the complexity of arbovirus transmission.

Elucidating the mechanisms of viral and vector interactions that determine the vector competence of insect species such as mosquitoes is critical to understand the dynamics of transmission of these viruses among vector populations and their transmission to human hosts. This manuscript by Crespo and colleagues addressed some biological aspects of the arbovirus infections of highly anthropogenic and competent mosquito species, Aedes aegypti and Culex quinquefasciatus from an area with high transmission for ZIKV and CHIKV such as the case of Brazil. The manuscript is well-written and the results well-described; however, these are still some aspects that needed to be clarified by the authors as noticed below:

Major(s)

1. Knowing that the vertical transmission of arboviruses such as dengue virus (DENV), ZIKV and CHIKV may occur in endemic areas; were natural isolated colonies of Ae. aegypti previously tested for DENV infection? This is important to know as pre-established viral infections in mosquito populations may affect their vectorial capacity once evaluated under laboratory conditions? RT-PCR? Was any other marker of arboviral infection is mosquitoes used?

R: Yes, It is a protocol in our lab to test pools of mosquitoes collected in the field for arbovirus and Wolbachia before establishing our colonies. So far, we have observed that vertical transmission is a very rare event and we found no positive sample for that colony. Besides, we did not use the F0 generation we have used F2.

2. Knowing the maturation stage that occur in flavivirus virions such as ZIKV and DENV once grown in either mammalian (e.g., Vero) or insect cells (C6/36 cells), was any experiment performed using viruses derived from infected mosquito cells?

R: No. We chose to use only virus grown in Vero cells, considering that a clear cytopathic effect is observed after infection in this particular type of cell.

3. For viral infection of both mosquito species, the authors used as negative control a mixture of equal volume of virus free cell culture and defibrinated rabbit blood; however, knowing that viral replication after infection of mosquitos would be the main cause of the altered biological parameters in mosquitoes, would not be better using inactivated viruses during blood feeding experiments instead?

R: We are not sure that viral replication is the main cause, or only the exposure to viruses after blood ingestion, without necessarily having infection and consequent replication (Styer, Meola and Kramer, 2007). Thus, with regard to the control, we chose to use blood free of viral particles. 

4. In the methods section the authors stated that “Whole mosquitoes were processed, except for Ae. aegypti collected at 7 dpe, whose abdomens and thorax were analyzed separately from the heads. How is that these important data was not included at least as a supporting information in this manuscript? This is a critical biological parameter for measuring vector competence and the level of viral infection (e.g., viral RNA load, viral genome equivalents, etc) detected in each individual mosquito that may affect the conclusions described in the manuscript. Please clarify this and add this set of data to the manuscript.

R: Thanks for this. We agree with reviewer, and we have performed a new analysis of correlation using this information and added to the supporting information. 

Reviewer #2: Peer Review for: Manuscript Number: PONE-D-23-02967 Manuscript Title: Exposure to Zika and chikungunya viruses impacts aspects of the vectorial capacity of Aedes aegypti and Culex quinquefasciatus

In their manuscript Crespo et al explore the impact of ZIKV and CHIKV arbovirus exposure and infection on reproductive capacity, longevity, and blood feeding of Aedes aegypti and Culex quinquefasciatus mosquitoes. They accomplish this by exposing lab and field-caught colonies of Ae. aegypti to both ZIKV and CHIKV, and a lab colony of Cx. quinquefasciatus to ZIKV and then assessing fecundity and fertility. To evaluate survival and longevity, the authors collected Ae. aegypti mosquitoes at 7, 14, and 21 days post exposure, but only collected Cx. quinquefasciatus when death occurred throughout the experiment. FTA cards infused with Manuka honey blend were utilized for viral detection. Additionally, blood-seeking behavior was assessed at 7, 14, and 21 days post exposure. RT-qPCR was performed for viral detection of the samples. While it’s significant that both viruses were associated with a biological cost, in particular, diminished reproductive capability, the fact that blood feeding activity remained the same (or even increased) after exposure helps further explain viral dissemination and global expansion of these arboviruses.

Major Issues:

1 - Line 170-171 - Why were Cx. quinquefasciatus mosquitoes treated differently (evaluated differently) from Ae. aegypti?

R: Data from experiments carried out by our group show a lower rate of ZIKV infection in Cx. quinquefasciatus when compared to Ae. aegypti. Thus, with the purpose of ensuring a minimum number of infected females for longevity assessment, we decided to collect the samples as the mosquitoes died, instead of collecting them at the three time points (7, 14 and 21 days). We clarify that the main objective of this study was to evaluate the biological cost after exposure and infection to the viruses and not to compare this cost between the evaluated species.

2 - Line 244-247 - More information is needed concerning the contamination of samples and inability to carry out 2nd and 3rd experiments. Did you consider re-running the experiment for the sake of replication?

R: Contamination was detected by RT-qPCR. We have changed the sentence in the manuscript. Considering that each experiment, in this case, takes around 60 days to finish and that the main objective of this study was to evaluate the post-exposure and infection biological cost and not the rate of transmission, we chose to proceed with the assessments with the data we had available and not repeat the experiments.

3 - Figures 1-3 are not visible on the downloaded PDF.

R: We have corrected it.

Minor Issues:

Please see my edited version of your manuscript with highlights and comments.

4 - Line 28 – no period is needed after “AeCamp”. 

R Done.

5 - Line 38-39 – consider defining “vectorial capacity” a bit here. You may also consider differentiating “vectorial capacity” from “vector competence”. 

R Done.

6 - Line 41-43 – The cited study involves Anopheles mosquitoes and doesn’t relate well as an “example” of what you previously stated.

R The citation was removed from the text.

7 - Line 67 – “vector” should read “vectorial”. 

R Done.

8 - Line 68-70 – consider re-wording this sentence as it is hard to understand.

R Done.

9 - Line 71 – use commas instead of semicolons. 

R Done.

10 - Line 83-84 – This could use a reference. 

R The citation was included in the text.

11 - Line 100 – What is “shelling”? 

R This was wrong, we have corrected it in the text. 

12 - Line 113 – “tittered” should be “titered”. 

R Done.

13 - Line 181 – Consider including the timeframe that mosquitoes were allowed to seek a blood meal. 

R Thank you. We have included it.

14 - For Figure captions, please include a clear description of what group each panel represents. For example, you do this in Fig 5 caption (A: RecL Colony; B: AeCamp Colony), but not clearly for Fig 4. 

R Done.

15 - Line 359 – Consider “blood feeding” instead of “blood meal activity”

Line 377 – remove the “a”. 

R Done.

16 - Line 494 – Consider changing “of control of” to “to control”. 

R Done.

Overall, this is a well designed and executed study to further explore the biological cost of arbovirus exposure and the impact of infection on vectorial capacity. Recommend acceptance for publication after minor revisions.

Reviewer #3: The paper discusses such an important issue, the effects of arbovirus on the biology of two insect vectors, Aedes aegypti, and Culex quinquefasciatus. Despite the interesting question, some details should be clarified.

It was challenging to analyze the results as the materials and methods needed to be clarified. It seems the authors made lots of exceptions in the experimental designs. If it is not a description problem, the results are not comparable because they were not made the same way.

Introduction

1 - Lines 41 – 43: “For example, the number of eggs produced in the first gonotrophic cycle indicates the total lay profile during the entire female life in mosquitoes [10].” I think some more recent references do not confirm this statement. Can you check this information?

R The example was used to justify that the assessment of the first gonadotrophic cycle is relevant, as it represents the profile of the female throughout her life. However, we agree that it is out of context. We have deleted from the text.

Materials and methods

2 - Lines 116 – 117: “For Ae. aegypti, each experiment was performed with two groups for each colony (RecL, 117 AeCamp, and CqSLab)”. CqSLab is a colony of C. quinquesfasciatus, isn’t it? I presume you meant one of the viruses, right? This point needs to be clarified.

R We meant that “each experiment was carried out with two groups for each colony (RecL, AeCamp and CqSLab): exposed to the virus (E) and not exposed (NE, control group)”. We have corrected in the text.

3 - Lines 133 – 134: How did you group the exposed but not infected (E) samples for ZIKV? Did you check all the mosquitoes individually to split them into the E and EI groups? This point needs to be clarified.

R Yes, we did run RT-qPCR for all mosquitoes individually and then separated those that were negative (exposed but not infected - E) and positive (exposed infected - EI). The information has been included in the text.

4 - Lines 136 – 140: Did you discard the E group for CHIKV or join them with the EI group?

R We have discarded group E, as the number of individuals was too small (the infection rate was too high – around 90%). The information has been included in the text.

5 - Lines 170 – 174: Survival and longevity assessment in Culex – why didn’t you repeat the same experimental design as Aedes? Why didn’t you collect samples at 7, 14, and 21 DPE?

R Plea

---

## [Decision Letter · Decision Letter 1]

2 Nov 2023

PONE-D-23-02967R1Exposure to Zika and chikungunya viruses impacts aspects of the vectorial capacity of Aedes aegypti and Culex quinquefasciatusPLOS ONE

Dear Dr. Ayres,

Thank you for submitting your manuscript to PLOS ONE. After careful consideration, we feel that it has merit but does not fully meet PLOS ONE’s publication criteria as it currently stands. Therefore, we invite you to submit a revised version of the manuscript that addresses the points raised during the review process.

The reviewers made points that need to be clarified or expanded upon to ensure your conclusions have the appropriate support.

Please submit your revised manuscript by Dec 17 2023 11:59PM If you will need more time than this to complete your revisions, please reply to this message or contact the journal office at plosone@plos.org. Please include the following items when submitting your revised manuscript:A rebuttal letter that responds to each point raised by the academic editor and reviewer(s). You should upload this letter as a separate file labeled 'Response to Reviewers'.A marked-up copy of your manuscript that highlights changes made to the original version. You should upload this as a separate file labeled 'Revised Manuscript with Track Changes'.An unmarked version of your revised paper without tracked changes. You should upload this as a separate file labeled 'Manuscript'.If applicable, we recommend that you deposit your laboratory protocols in protocols.io to enhance the reproducibility of your results. Protocols.io assigns your protocol its own identifier (DOI) so that it can be cited independently in the future. For instructions see: https://journals.plos.org/plosone/s/submission-guidelines#loc-laboratory-protocols. Additionally, PLOS ONE offers an option for publishing peer-reviewed Lab Protocol articles, which describe protocols hosted on protocols.io. Read more information on sharing protocols at https://plos.org/protocols?utm_medium=editorial-email&utm_source=authorletters&utm_campaign=protocols.

We look forward to receiving your revised manuscript.

Kind regards,

Kelli L. Barr, Ph.D.

Academic Editor

PLOS ONE

Journal Requirements:

Reviewers' comments:

Reviewer's Responses to Questions

**Comments to the Author**

1. If the authors have adequately addressed your comments raised in a previous round of review and you feel that this manuscript is now acceptable for publication, you may indicate that here to bypass the “Comments to the Author” section, enter your conflict of interest statement in the “Confidential to Editor” section, and submit your "Accept" recommendation.

Reviewer #1: All comments have been addressed

Reviewer #2: All comments have been addressed

Reviewer #3: (No Response)

2. Is the manuscript technically sound, and do the data support the conclusions?

Reviewer #1: Partly

Reviewer #2: Yes

Reviewer #3: Yes

3. Has the statistical analysis been performed appropriately and rigorously? 

Reviewer #1: Yes

Reviewer #2: Yes

Reviewer #3: Yes

4. Have the authors made all data underlying the findings in their manuscript fully available?

Reviewer #1: Yes

Reviewer #2: Yes

Reviewer #3: Yes

5. Is the manuscript presented in an intelligible fashion and written in standard English?

Reviewer #1: Yes

Reviewer #2: Yes

Reviewer #3: No

6. Review Comments to the Author

Reviewer #1: (No Response)

Reviewer #2: Review for Revision 1 of:

Manuscript Number: PONE-D-23-02967

Manuscript Title: Exposure to Zika and chikungunya viruses impacts aspects of the vectorial capacity of Aedes aegypti and Culex quinquefasciatus

Additional Minor Issues:

Line 50 – Add space before “Modifications”

Line 54 – Flaviviridae and Togaviridae should both be italicized ( Flaviviridae and Togaviridae )

Line 88 – Add a period.

In the “Study Area” section, consider adding avg. annual precipitation.

Line 92 – remove “and”

Line 102 – delete duplicate “and”

Line 103 – delete space and 1 “.” after the word “dipper”

Line 167 – use a comma, not a semicolon as in line 173.

Line 231 – no ‘ needed after “Wallis”

Table 2 is difficult to interpret; data doesn’t seem to fit correctly. Be consistent. Some numbers use “e” others “E” and 2 of the numbers just have a long string of integers.

Consider just using decimal points for the numbers in your tables, not commas.

Line 359 – 360 – delete the space between 63.48 and the “%” as well as with 40.67

Line 363 – add % after 57.50 and 37.50

Line 400 – delete extra comma after “6”

Line 460 – I believe the word you intend, throughout the paper is “gonotrophic” as opposed to gonadotropic or gonadotrophic. Consider using gonotrophic at line 134, 157, 162, 168, Fig 4 legend, 370, and 460.

Line 540 – It is worth highlighting…

Line 543 – “pointed out” of better, “discussed” or “elucidated”

Reviewer #3: The manuscript has improved after the suggestions made by the reviewers, and I thank the authors for that. However, I still have some considerations about the data and the conclusions made.

The authors wrote in the Discussion section: “In summary, despite the significant reduction in some aspects of the biological performance of Ae. aegypti, for both viruses and for Cx. quinquefasciatus, infected with ZIKV, our data suggest that their vectorial capacity is probably not affected, as the longevity and feeding behavior of field mosquitoes were not impacted by virus.” They also mention that the data presented in the manuscript corroborated other published data. Were the authors expecting any different results? Why or why not? Would you expect to find any differences in mosquitoes from the field? If so, do you have any hypothesis to explain why it did not happen? Sorry if I am being repetitive, but I still miss some sentences describing the new relevant data of this manuscript.

7. PLOS authors have the option to publish the peer review history of their article (what does this mean?). If published, this will include your full peer review and any attached files.

Reviewer #1: **Yes: **Henry Puerta-Guardo

Reviewer #2: No

Reviewer #3: No

---

## [Author Response · Author response to Decision Letter 1]

19 Dec 2023

Review Comments to the Author

Reviewer #2: Review for Revision 1 of:

Manuscript Number: PONE-D-23-02967

Manuscript Title: Exposure to Zika and chikungunya viruses impacts aspects of the vectorial capacity of Aedes aegypti and Culex quinquefasciatus

Additional Minor Issues:

Line 50 – Add space before “Modifications”

Author's response to reviewer: done

Line 54 – Flaviviridae and Togaviridae should both be italicized (Flaviviridae and Togaviridae )

Author's response to reviewer: done

Line 88 – Add a period.

Author's response to reviewer: done

In the “Study Area” section, consider adding avg. annual precipitation.

Author's response to reviewer: done

Line 92 – remove “and”

Author's response to reviewer: done

Line 102 – delete duplicate “and”

Author's response to reviewer: done

Line 103 – delete space and 1 “.” after the word “dipper”

Author's response to reviewer: done

Line 167 – use a comma, not a semicolon as in line 173.

Author's response to reviewer: done

Line 231 – no ‘ needed after “Wallis”

Author's response to reviewer: done

Table 2 is difficult to interpret; data doesn’t seem to fit correctly. Be consistent. Some numbers use “e” others “E” and 2 of the numbers just have a long string of integers.

Consider just using decimal points for the numbers in your tables, not commas.

Author's response to reviewer: done

Line 359 – 360 – delete the space between 63.48 and the “%” as well as with 40.67

Author's response to reviewer: done

Line 363 – add % after 57.50 and 37.50

Author's response to reviewer: done

Line 400 – delete extra comma after “6”

Author's response to reviewer: done

Line 460 – I believe the word you intend, throughout the paper is “gonotrophic” as opposed to gonadotropic or gonadotrophic. Consider using gonotrophic at line 134, 157, 162, 168, Fig 4 legend, 370, and 460.

Author's response to reviewer: done

Line 540 – It is worth highlighting…

Author's response to reviewer: done

Line 543 – “pointed out” of better, “discussed” or “elucidated”

Author's response to reviewer: done 

Reviewer #3: 

The manuscript has improved after the suggestions made by the reviewers, and I thank the authors for that. However, I still have some considerations about the data and the conclusions made.

The authors wrote in the Discussion section: “In summary, despite the significant reduction in some aspects of the biological performance of Ae. aegypti, for both viruses and for Cx. quinquefasciatus, infected with ZIKV, our data suggest that their vectorial capacity is probably not affected, as the longevity and feeding behavior of field mosquitoes were not impacted by virus.” They also mention that the data presented in the manuscript corroborated other published data. Were the authors expecting any different results? Why or why not? Would you expect to find any differences in mosquitoes from the field? If so, do you have any hypothesis to explain why it did not happen? Sorry if I am being repetitive, but I still miss some sentences describing the new relevant data of this manuscript. 

Author's response to reviewer: We changed the last paragraph of the discussion to clarify these issues.

---

## [Editor Report · Decision Letter 2]

2 Jan 2024

PONE-D-23-02967R2Exposure to Zika and chikungunya viruses impacts aspects of the vectorial capacity of Aedes aegypti and Culex quinquefasciatusPLOS ONE

Dear Dr. Ayres,

Thank you for submitting your manuscript to PLOS ONE. After careful consideration, we feel that it has merit but does not fully meet PLOS ONE’s publication criteria as it currently stands. Therefore, we invite you to submit a revised version of the manuscript that addresses the points raised during the review process.

Your paper is much improved but there are a few issues that require clarification. Reviewer 2 identified minor grammatical and formatting errors that must be addressed.  Reviewer 3 raised questions regarding your conclusions. Addressing these questions will support your conclusions.

We look forward to receiving your revised manuscript.

Kind regards,

Kelli L. Barr, Ph.D.

Academic Editor

PLOS ONE
---

## [Author Response · Author response to Decision Letter 2]

5 Apr 2024

Dear Editor

As I mentioned before, we did not receive the comments from the reviewers, so we are submitting the files again.

---

## [Editor Report · Decision Letter 3]

15 Apr 2024

Exposure to Zika and chikungunya viruses impacts aspects of the vectorial capacity of Aedes aegypti and Culex quinquefasciatus

PONE-D-23-02967R3

Dear Dr. Ayres,

We’re pleased to inform you that your manuscript has been judged scientifically suitable for publication and will be formally accepted for publication once it meets all outstanding technical requirements.

Kind regards,

Kelli L. Barr, Ph.D.

Academic Editor

PLOS ONE